# On the Study of Influences of Different Factors on the Rapid Tropospheric Tomography

**Wenxuan Liu [1], Yidong Lou [1,2], Weixing Zhang [1,*], Jinfang Huang [1], Yaozong Zhou [1] and Haoshan Zhang [3]**

[1]  GNSS Research Center, Wuhan University, Wuhan 430079, China
[2]  Collaborative Innovation Center of Geospatial Technology, Wuhan University, Wuhan 430072, China
[3]  State Key Laboratory of Information Engineering in Surveying, Mapping and Remote Sensing, Wuhan University, Wuhan 430079, China
[*]  Correspondence: zhangweixing89@whu.edu.cn

**Abstract:** A rapid tropospheric tomography system was developed by using algebraic reconstruction technique. Influences of different factors on the tomographic results, including the ground meteorological data, the multi-Global Navigation Satellite System (GNSS) observations, the ground station distribution and the tomographic horizontal resolution, were systematically investigated. In order to exclude the impacts from discrepancies of water vapor information between input observations and references on the tomographic results, the latest reanalysis products, ERA5, which were taken as references for result evaluations, were used to simulate slant wet delay (SWD) observations at GNSS stations. Besides, the slant delays derived from GNSS processing were also used to evaluate the reliability of simulated observations. Tomography results show that the input both SWD and ground meteorological data could improve the tomographic results where SWD mainly improve the results at middle layers (500 to 5000 m, namely 2 to 16 layer) and ground meteorological data could improve the humidity fields at bottom layers further (0 to 500 m, namely 0 to 2 layer). Compared to the usage of Global Positioning System (GPS) only SWD, the inclusion of multi-GNSS SWD does not significantly improve the tomographic results at all layers due to the almost unchanged dispersion of puncture points of GNSS signals. However, increases in the ground GNSS stations can benefit the tomography, with improvements of more than 10% at bottom and middle layers. Higher tomographic horizontal resolution can further slightly improve the tomographic results (about 3-6% from 0.5 to 0.25 degrees), which, however, will also increase the computational burden at the same time.

**Keywords:** water vapor; tomography; simulation; GNSS

## 1. Introduction

As the most abundant greenhouse gas, atmospheric water vapor plays an important role in the climate and weather system. Due to the complex variations of water vapor in both spatial and temporal domains, measuring atmospheric water vapor contents precisely is still challenging. Compared to other water vapor measuring tools, the ground-based Global Navigation Satellite System (GNSS) has the advantages of high-accuracy, high temporal resolution, continuous and all-weather operation, which makes ground-based GNSS become a powerful technique for atmosphere sounding [1]. The relatively low temporal resolution of water vapor measurements in the operational system (e.g., radiosonde) is one of the main reasons for the poor forecasting of some extreme weather events (e.g., severe storm). Applications of ground-based GNSS water vapor measurements are therefore very promising in extreme weather forecasting [2,3]. In addition, signal delays caused by atmospheric water

vapor is also one of the main errors in geodetic data analysis such as GNSS [1,4,5], very long base (VLBI) [6,7], and satellite laser ranging (SLR) [7,8]. Information about the water vapor content and variations can therefore be used to correct GNSS and other measurements.

Traditionally, in the GNSS data analysis, the Zenith Tropospheric Delay (ZTD) or the Precipitable Water Vapor (PWV) is derived, which represent the vertical integrated tropospheric delays or water vapor content over the GNSS receiver, but information about the vertical distribution of troposphere remains unknown. However, the vertical distribution of humidity fields can be valuable in some applications, such as the study of the convection system construction [9–11].

Bevis et al. [1] proposed the idea of using the ground-based GNSS network to reconstruct the three-dimensional (3D) distribution of water vapor in atmosphere, namely, the tropospheric tomography. The first tropospheric tomography experiment was carried out by Flores et al. [12]. After that, many campaigns or experiments have been conducted over different regions to illustrate the potential of GNSS tomography to provide spatially resolved humidity fields [13–15]. However, due to the geometric distribution of the ground-based GNSS station and the geometry of the satellite constellation, some voxels in the tomographic region cannot be touched by any GNSS signal ray paths, resulting in a rank-deficient issue in the tropospheric tomography. This issue can be solved by introducing additional constraints, for example, by adding the vertical constraints [16], or taking numerical weather prediction (NWP) model as initial values [17]. Another challenging issue in the tomography is how to efficiently estimate the massive parameters, especially when the scale of the tomographic region grows or in (near) real-time applications. Most of previous troposphere tomography studies aimed at small regions, generally in the scale of several tens of kilometers with relatively dense GNSS networks [12,18]. There have been few attentions paid to rapid tomography. However, rapid tropospheric tomography can be very useful in some applications, such as providing timely and complete pictures for some extreme weather phenomena, supporting the development of high-resolution mapping functions, and providing consistent tropospheric delay corrections for (near) real-time precise navigation or positioning users.

Tomographic results can be affected by many factors. For example, Yu et al. [19] compared various constraint conditions. As results show, with the water vapor changing gently in horizontal direction, the weight matrix of horizontal constraints has a large effect on the tomographic result. Xia et al. [20] combined Global Positioning System (GPS) and Global Navigation Satellite System (GLONASS) precise point positioning (PPP) to obtain the three-dimensional atmospheric water vapor distribution. Test results indicate that the number of the voxels from GPS/GLONASS PPP that are passed by the satellite rays increased by 18%, and the water vapor accuracy from GPS/GLONASS PPP is 10% higher than with the GPS PPP method. Results in Dong et al. [21] show that the integrated multi-GNSS can pronouncedly increase the number of effective signals, and three-dimensional water vapor results are better than those from the GPS-only system, improving by 5% with GPS + GLONASS or GPS + GLONASS + BeiDou Navigation Satellite System (BDS). Notarpietro et al. [22] found that the tomographic horizontal resolution can be improved by densifying the GNSS network, and the vertical resolution can be improved if receivers are deployed at different altitudes. However, in most current studies, the radiosonde or the reanalysis products are used as references to evaluate the tomographic results [16,18]. This kind of evaluation may be affected by the discrepancy among different techniques, for example, the differences of water vapor information between GNSS and radiosonde or reanalysis products rather than the tomography itself [23,24].

In this work, we will utilize the reanalysis products to simulate GNSS slant wet delays which will then be used as inputs in the tomography. The wet refractivity filed derived from the tomography is evaluated by comparing with reanalysis products, ERA5 (as a true value), to exclude the influence of differences between inputs and references. Influences of different factors including the ground meteorological observation data, the multi-GNSS observations, the ground-based station distribution and the horizontal resolution on the tomographic results, are discussed and analyzed. The NWP forecast products are used to avoid the rank-deficiency and an algebraic reconstruction technique is

applied to tackle the massive parameter estimation issue in large-scale region tomography. The data used in this study is introduced in Section 2. The tomographic method is described in Section 3. The slant wet delay (SWD) simulation based on reanalysis products is depicted in Section 4, and the tomography experiments and result discussion are presented in Section 5, followed by the conclusions in Section 6.

## 2. Data

Data used in this study is introduced in this section, including the GNSS data, the reanalysis product and the NWP forecast product.

### 2.1. GNSS Data

The tomography region in this work will mainly cover the Guangdong Province and the surrounding area in the southern China as shown in Figure 1. Observations from GPS and GLONASS at 95 stations in Guangdong Continuously Operating Reference Stations (CORS) network with the sampling rate of 30 s on April 29 (day 119) 2017 is processed. Since this study will make an effort to exclude the influences of the discrepancy in SWD between the GNSS and the reference (reanalysis product), we will not use the GNSS-derived SWD as inputs in the tomography experiments, but only use GNSS to make comparisons with the reanalysis-derived slant delays.

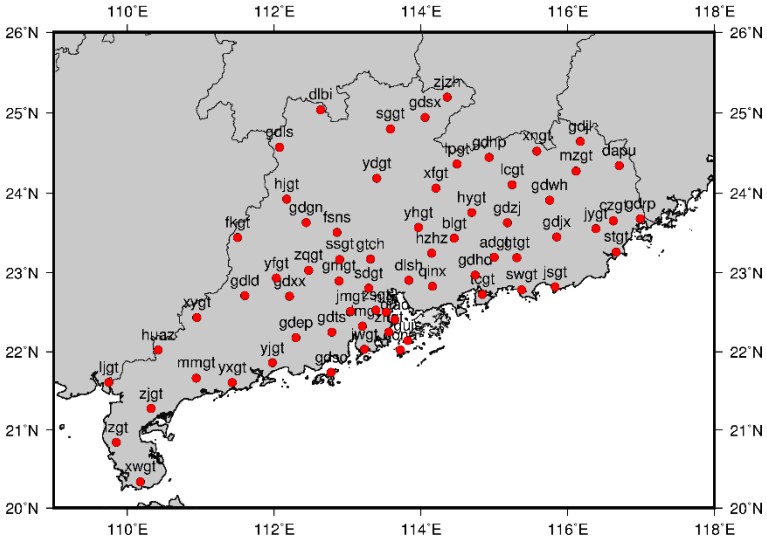

**Figure 1.** Distribution Map of Continuously Operating Reference Stations (CORS) Station in Guangdong Province.

### 2.2. Reanalysis Product

Meteorological reanalysis product provides comprehensive global multi-decadal records of historical atmosphere states using a single consistent numerical data assimilation scheme with various past observations. The latest reanalysis, ERA5, produced by European Centre for Medium-Range Weather Forecasts (ECMWF) with a temporal resolution of 1 h and a horizontal resolution of 0.25 degrees is used to simulate the SWD and meteorological parameters (i.e., temperature, pressure and relative humidity) at GNSS stations. In addition, the reanalysis product will also be utilized to calculate the wet refractivity by Equation (1) that is taken as the reference for tomographic result evaluations.

$$N_w = Z_w^{-1}\left(k_2\frac{e}{T} + k_3\frac{e}{T^2}\right) \tag{1}$$

where $e$ is the partial pressure of water vapor, $T$ is the temperature and $Z_w^{-1}$ is the inverse compressibility factor of water vapor (in general, $Z_w^{-1} = 1$). The constant $k_2 = 70.4$ K hpa$^{-1}$ and $k_3 = 3.739 \times 10^5$ K$^2$ hpa$^{-1}$.

*2.3. NWP Forecast Product*

The Global Forecast System (GFS) is a weather forecast model produced by the National Centers for Environmental Prediction (NCEP). Dozens of atmospheric and land-soil variables are available through this dataset, from temperatures, winds, and precipitation to soil moisture and atmospheric ozone concentration. The entire globe is covered by the GFS at a base horizontal resolution of 18 miles (28 kilometers) between grid points. The wet refractivity estimated by Equation (1) from GFS forecast product is taken as initial values in tomography.

## 3. GNSS Tropospheric Tomography Methodology

*3.1. Tomography Mathematic Model*

The GNSS signal transmitted from satellite to ground receiver can be delayed by the ionosphere and neutral atmosphere. Ionospheric delay can be minimized by combinations of observations at different frequencies. Most of the neutral atmospheric delays are induced by troposphere, so the neutral atmospheric signal delay is generally referred as tropospheric delay. In GNSS data processing, tropospheric delay along the signal path is generally mapped into the zenith direction, namely the zenith tropospheric delay (ZTD), to avoid a rank deficiency problem [25].

ZTD can be divided into zenith hydrostatic delay (ZHD) and zenith wet delay (ZWD) component as:

$$ZTD = ZWD + ZHD \tag{2}$$

ZHD can be accurately estimated by empirical models, such as the Hopfield or Saastamoinen model [26,27]. ZWD, which is mainly caused by atmospheric water vapor, however, is hard to be accurately modeled. In GNSS meteorology, ZWD is generally obtained by subtracting ZHD from the estimated ZTD, the by-product in GNSS data processing, namely,

$$ZWD = ZTD - ZHD. \tag{3}$$

As mentioned previously, the ZTD or ZWD does not contain vertical distribution of humidity. In tropospheric tomography, the zenith delays need to be converted to slant delays along the GNSS signal path as inputs in tomography. The slant tropospheric delay (STD) can be calculated according to,

$$STD = SHD + SWD = m_h \cdot ZHD + SWD \tag{4}$$

where $m_h$ is the hydrostatic mapping function.

Regarding the slant wet delay (SWD), with the estimated ZWD from Equation (3), horizontal gradients and the post-fit residuals from GNSS data processing, the slant wet delays (SWD) can be retrieved according to [28],

$$SWD = m_w \cdot [ZWD + cot\varepsilon (G_N cos\phi + G_E sin\phi)] + \delta \tag{5}$$

where $m_w$ is the wet mapping function, $G_N$ and $G_E$ are the delay gradient parameters in the north-south and east-west direction, respectively, $\varepsilon$ denotes the elevation angle, $\phi$ is the azimuth, and $\delta$ is the post-fit phase residual.

SWD is actually the integral of the wet refractivity, $N_w$, along the signal path ($S$), as following,

$$SWD = 10^{-6} \int_s N_w d_s \tag{6}$$

In tomography, we generally discretize the troposphere into 3D voxels and $N_w$ in Equation (6) then denotes the wet refractivity in each voxel. After we get the SWD from Equation (5), Equation (6) can be discretized and written as,

$$\mathbf{SWD} = \mathbf{AX} \tag{7}$$

where **A** is the projection matrix where each element denotes the distances of the signal that passes the voxel, and the vector **X** contains the to-be-estimated wet refractivity in each voxel.

### 3.2. Tomography Parameter Estimation Method

Rapid reconstructions of water vapor fields are meaning for providing high temporal and spatial humidity fields and improving the accuracy of numerical weather forecast, et al. Therefore, how to estimate the parameters efficiently in large-scale region tomography is of great importance, especially for near real-time applications. The most commonly used parameter estimation methods like the least squares and Kalman filter need the operation of matrix inversion. As the scale of tomography region grows, the size of the matrix can be significantly huge which will make the inversion very time-consuming. In this study, the algebraic reconstruction technique (ART) is utilized. The algebraic reconstruction technique iterates over the initial estimate until a certain condition is satisfied. ART avoids matrix inversion, which is especially suitable for operations involving large dimensional matrices, such as the tropospheric tomography in our case.

ART is a frequently used image reconstruction algorithm. The technique can also be applied to the tomography. The ART family mainly include the original additive algebraic reconstruction techniques, the multiplicative algebraic reconstruction techniques (MART) and the simultaneous iterations reconstruction technique (SIRT). The algebraic reconstruction techniques have been successfully used to solve the computational efficiency issue in ionospheric tomography in some previous studies. For example, Stolle et al. [29] and Jin et al. [30] applied the multiplicative algebraic reconstruction to ionospheric tomography, Das et al. [31] compared the ART and MART, Wen et al. [32] presented an improved algebraic reconstruction technique (IART), and Yao et al. [33] proposed adaptive simultaneous iteration reconstruction technique (ASIRT) on the basis of simultaneous iteration reconstruction technique.

The ART algorithm assumes a linear relationship between the values to be reconstructed and the projected values for each grid element, and the grid parameters are corrected one by one during each iteration as,

$$x_j^{k+1} = x_j^k + \lambda a_{ij} \frac{SWD_i - \sum_{j=1}^n a_{ij} x_j^k}{\sum_{j=1}^n a_{ij}^2} \tag{8}$$

where $x$ is the wet refractivity in the grid; $k$ is the number of iteration; $n$ is the number of columns of the observation equation, namely the number of parameters; $a_{ij}$ is the element in the $i$ row and $j$ column of the equation coefficient matrix; $\lambda$ is the relaxation parameter; $SWD$ is the observed value.

The result of the ART depends on the order of the data $SWD_i$ as shown in Equation (8). To avoid this behavior, SIRT no longer individually modifies the water vapor density on each observation path, but iterates and modifies all the observed paths at one time as,

$$x_j^{k+1} = x_j^k + \lambda \sum_{i=1}^m \frac{a_{ij}\left(SWD_i - \sum_{j=1}^n a_{ij} x_j^k\right)}{\sum_{j=1}^n a_{ij}^2} \tag{9}$$

where $m$ is the number of rows of the observation equation (including the constraint equation).

In order to achieve faster convergence and better estimates compared to the plain ART, an adaptive SIRT (ASIRT) with a data-driven adjustment of the weight has been applied in some ionosphere tomographic studies (e.g., Yao et al.) [33]. The ASIRT formula can be written as,

$$x_j^{k+1} = x_j^k + \lambda \sum_{i=1}^{m} \frac{a_{ij} x_j^k \left( SWD_i - \sum_{j=1}^{n} a_{ij} x_j^k \right)}{\sum_{j=1}^{n} a_{ij}^2 x_j^k}. \tag{10}$$

Compared to Equation (9), estimates from the previous step are taken into account in the determination of the weight in ASIRT.

### 3.2.1. Relaxation Parameter and Iterative Termination Criteria

The relaxation parameter $\lambda$ acts as a weight in the iterative formula, and adjusts the magnitude of the correction for each iteration. If the relaxation parameter is too large, the correction will be large and the iteration will be prone to get divergence. On the contrary, small relaxation parameter may result in slow convergence. In this study, the relaxation parameter is set to be 0.015 after a series of trials.

The ART algorithm requires multiple iterations to reach convergence and the criteria that describes the convergence behavior to terminate the iteration in an optimal manner needs to be pre-defined. However, there is no uniform criterion, and in this paper, the consistency between the estimates and observations is used define the iterative termination criteria.

The $k$-th computed value $SWD^k$ can be obtained by taking the estimated parameters back to the observation equation, and the difference between the computed and the observed value can be expressed as:

$$dSWD^k = SWD^k - SWD. \tag{11}$$

The root mean square (*rms*) of $dSWD^k$ can be calculated as:

$$rms = \sqrt{\frac{1}{m} \sum_{i=1}^{m} dSWD_i^k} \tag{12}$$

where $m$ is the number of cycles in the $k$-th iteration. Iteration is terminated when $rms < 0.002$ or after 200 iterations.

### 3.2.2. Initial Value and Gaussian Smoothing Filter

Due to the geometry of the ground GNSS stations and the satellite constellation, the projection matrix is generally sparse and ill-conditioned. This issue can be solved by adding additional constraints, for example, introducing initial values in tomography equations [34]. The initial value can be obtained in several ways, for example, by using the standard atmosphere, by spatially interpolating the ground-based meteorological measurements, or by utilizing the NWP products. Reasonable initial values can not only solve the rank deficient issue, but also speed up the convergence and improve the quality of the tomographic results. In this work, the global forecast products (GFS) produced by National Centers for Environmental Prediction (NCEP), which can be accessed in (near) real time, is taken as initial value.

As mentioned previously, some voxels in tropospheric tomography cannot be touched by any GNSS signal due to the geometry of ground GNSS stations and satellite constellation. In addition, the signals passed out of the tomographic region from the side faces are generally removed, which makes the ill-condition issue worse. In tomography, by using the algebraic reconstruction technique as described in Section 2.2, Nw in voxels without signal passing through will remain unchanged, which may induce

unrealistic tropospheric fields with spurious values. The Gaussian filter will therefore be used in this work to reduce the influence of this issue [28]. The filter operator can be expressed as follows:

$$G(x, y) = \frac{1}{2\pi\sigma^2} e^{-\frac{s^2}{2\sigma^2}} \tag{13}$$

where *s* is the distance between the surrounding grid node and the center node, and *σ* represents the standard deviation. Two kinds of Gaussian template with order of five are used where *σ* = 1 and *σ* = 0.3 are used for voxel without and with GNSS signals passed, respectively.

## 4. Slant Wet Delay Simulations

In general, most current studies used the radiosondes or the reanalysis products to evaluate the tomographic results. However, the differences of water information between GNSS and radiosonde or reanalysis products may affect the evaluation besides the tomographic technique itself. In this work, the reanalysis products are utilized to simulate the GNSS slant wet delays as inputs in tomography, and the tomographic results are evaluated by comparing with reanalysis products to exclude the influence of differences between inputs and references.

### 4.1. Simulation Procedure

The wet refractivity field estimated from ERA5 product is used as the basis for SWD simulation. The precise multi-GNSS satellite orbit products (GBM) generated by the German Research Centre for Geosciences (GFZ) are utilized to calculate the satellite positions with an interval of 30 s, and the satellite azimuth and elevation angle can be estimated according to the ground-based GNSS station coordinate. The one-dimensional ray-tracing method is then applied to estimate the SWD (STD) from the ERA5 wet (total) refractivity field. The schematic diagram of STD and SWD simulation is presented in Figure 2.

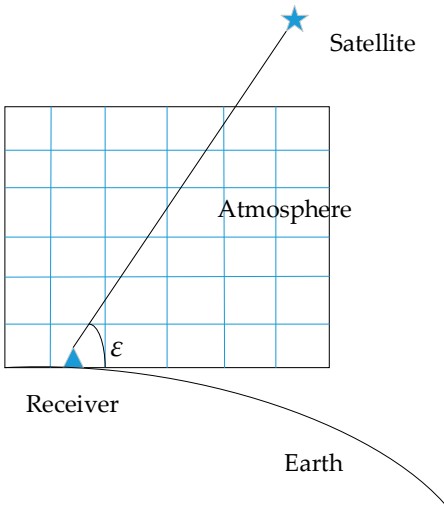

**Figure 2.** Schematic diagram of slant tropospheric delay (STD) and slant wet delay (SWD) simulation.

Firstly, the approximate signal path is determined according to the azimuth and elevation angle of the satellite. Then, SWD is obtained by piecewise integration of the ray in the tomographic region following Equation (6). Both the STD and SWD are simulated, where STD is compared with the GNSS-derived STD in Section 4.2 and SWD is taken as inputs in the tomography experiments.

### 4.2. Simulated Slant Tropospheric Delay Evaluation

Although we use the simulated measurements as inputs in the tomography system, and then use ERA5 to verify the accuracy of the tomographic results. In theory, the discrepancy between the simulated STD and GNSS-derived STD has no impacts on the tomographic results, but through this kind of comparisons, we can learn the accuracy of simulated SWD and STD based on ERA5, and provide a meaningful reference for other studies.

The procedure for simulated STD evaluation is shown in Figure 3. GNSS observations at stations in Guangdong Province in southern China as shown in Figure 1 are processed by PANDA software [35]. The ERPs (earth rotation parameters) and the DCBs (differential code biases) provided by CODE were employed. The absolute antenna phase center, phase windup corrections and station displacement corrections suggested in the IERS 2003 conventions were applied. The elevation cut-off angle was 10 and an elevation-dependent weighting strategy was applied to measurements at low elevations. The Global mapping function was used to relate zenith tropospheric delays (ZTDs) to the measurements. From a selected CORS network, the satellite clocks are estimated firstly as white noise with the satellite orbits and the station coordinates fixed, and then the satellite orbit and the estimated clocks are applied to user station for kinematic PPP in post-mission mode. ZTDs are estimated as piece-wise constants with a constraint of 4 cm$^2$ per hour and ambiguities are estimated as float solutions.

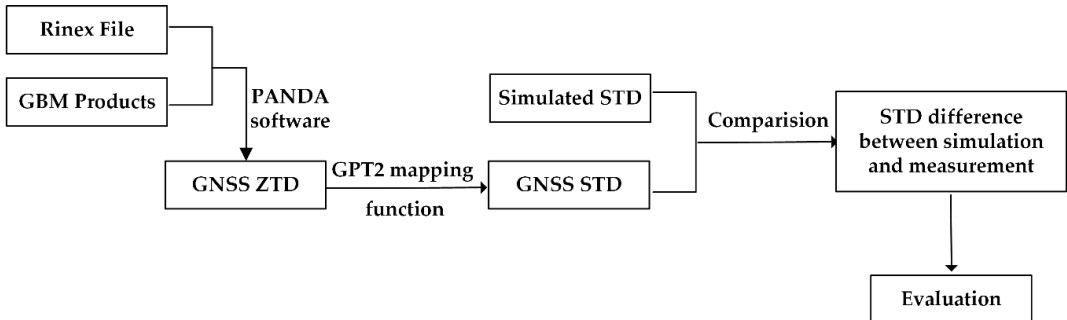

**Figure 3.** Flow chart of simulated STD evaluation.

These stations will also be used for tomography experiments in the next section. The generated ZTDs are then mapped into the signal path direction and differences between the simulated and GNSS STD are calculated. Variations of STD difference with satellite elevation for two GPS satellites at one station, BLGT, on day 119, 2017 are taken as an example as shown in Figure 4, where we can find that the difference is obviously smaller at higher satellite elevation angle. The difference can reach 15–20 cm at 10° elevation while is only about 2.5–5 cm at 60° elevation.

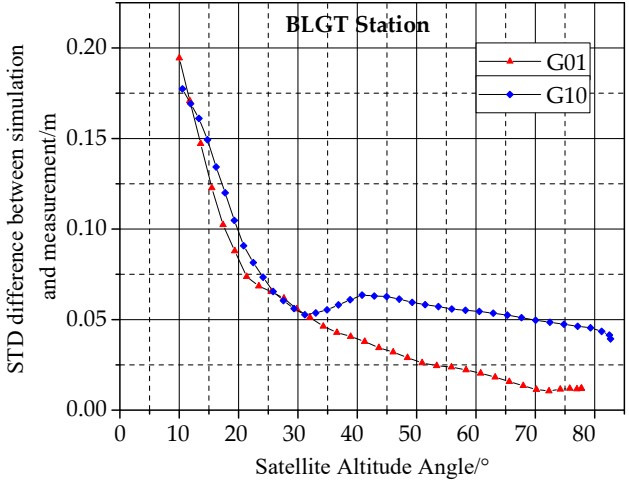

**Figure 4.** STD difference between simulation and measurement of G01 and G10 satellite at BLGT station.

RMS of STD differences for individual GPS satellites at all stations as well as the average RMS at different satellite elevation angles for all observations on day 119, 2017 are presented in Figure 5. We can find that the average RMS for individual satellite is generally between 0.03 to 0.06 m, and significantly decreases with the satellite elevation angles. For satellite with elevation below 10 degrees, the average RMS can exceed 0.5 m, compared to only about 0.02 m for satellites with elevation larger than 80 degrees.

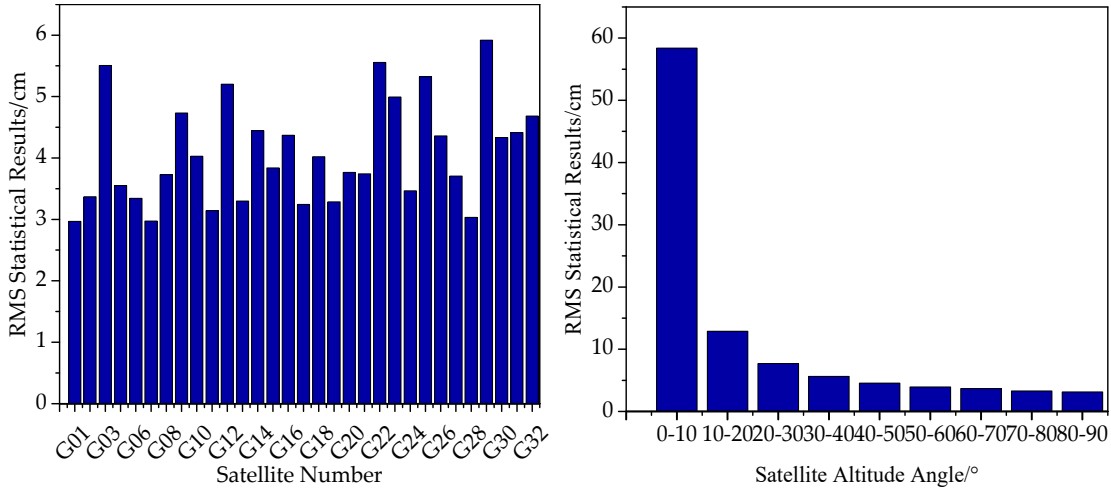

**Figure 5.** The root mean square (RMS) of the STD differences between simulated and Global Navigation Satellite System (GNSS) STD with satellite and satellite elevation angle.

## 5. Tomography Experiments and Results

SWD at 93 GNSS stations in Guangdong Province on the day 119 in 2017 and the meteorological observations (air temperature, pressure, and relative humidity) at GNSS station were simulated on the basis of ERA5. The tomographic initial values are from GFS forecast data. In this section, we will conduct tomographic experiments based on the simulated observations to study the influences of meteorological observation data, multi-GNSS observations, ground-based station distribution, and tomographic horizontal resolution on the tomographic results.

The tomography region covers the whole Guangdong Province and surrounding area in south China with latitude from 20° N to 25.5° N and longitude from 109° E to 118.5° E. In the control case (CTRL), the horizontal resolution is set to be 0.5 degrees and the vertical space is divided into layers with resolution of 300, 500, and 1000 m for height below 3 km, between 3 and 5 km, and between 5 and 12 km, respectively, resulting in $12 \times 20 \times 24 = 5760$ voxels in total. The tomographic interval is set to be 30 min.

Besides the control case, seven experiments (CS01–CS07) as described in Table 1 are setup to study the influences of different factors on tomographic results. In Table 1, "MET Obs" means meteorological observations, "Computational Time" means the time required to calculate half an hour data (The tomographic results are output every half an hour) (PC configuration: CPU, Intel Core i7 – 7700 @ 3.60 GHz, 3.60 GHz; RAM, 8G), "No" means the specific observations are not included in the tomography, "N/A" stands for not applicable, and "Measured" and "Simulated" mean that the GNSS station coordinates are from actual station coordinates or from simulation, respectively.

**Table 1.** Setup of tomographic experiments.

| Case Name | Short Name | MET Obs | GNSS Systems | Horizontal Resolution | Station Distribution | Number of Station | Number of Voxel | Computational Time |
|---|---|---|---|---|---|---|---|---|
| Control case | CTRL | No | No | 0.5° × 0.5° | N/A | N/A | 5760 | 1.5 s |
| Case 01 | CS01 | No | GPS | 0.5° × 0.5° | Real GNSS stations | 93 | 5760 | 7.5 s |
| Case 02 | CS02 | Yes | GPS | 0.5° × 0.5° | Real GNSS stations | 93 | 5760 | 8 s |
| Case 03 | CS03 | Yes | GPS + GLONASS + Galileo + BDS | 0.5° × 0.5° | Real GNSS stations | 93 | 5760 | 21 s |
| Case 04 | CS04 | Yes | GPS | 0.5° × 0.5° | Simulated 1.0° × 1.0° | 45 | 5760 | 5 s |
| Case 05 | CS05 | Yes | GPS | 0.5° × 0.5° | Simulated 0.5° × 0.5° | 153 | 5760 | 14 s |
| Case 06 | CS06 | Yes | GPS | 0.5° × 0.5° | Simulated 0.2° × 0.2° | 861 | 5760 | 47 s |
| Case 07 | CS07 | Yes | GPS | 0.25° × 0.25° | Simulated 0.2° × 0.2° | 861 | 21528 | 255 s |

### 5.1. Influences of Ground Meteorological Observation Data

In order to study the influences of adding ground meteorological data, three cases were carried out, namely, (1) only using GFS data as background field without input observation data (CTRL); (2) using GFS data as background field with GPS SWD as inputs (CS01); (3) using GFS data as background field with GPS observation data and ground meteorological observation data as inputs (CS02). All the tomography input data (SWD and meteorological observations) are simulated following procedures described in Section 3.1.

As mentioned previously, the ERA5 product, which is used as the basis for SWD and meteorological observation simulation, is again taken as the reference for tomographic result evaluation. The wet refractivity error profiles derived from CTRL, CS01 and CS02 are presented in Figure 6 where the horizontal axis denotes the vertical layer and the vertical axis is the standard deviation of wet refractive differences for all voxels with GNSS ray passed between tomographic outputs and ERA5-derived *Nw*.

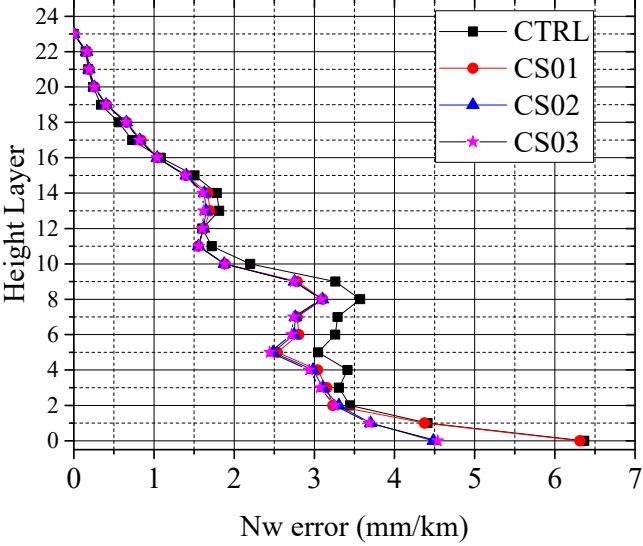

**Figure 6.** Comparisons of wet refractivity error profiles for CTRL, CS01, CS02, and CS03 for a study of influences of ground observations and ground multi-GNSS observations on tomography.

It can be seen from Figure 6 that *Nw* error at the lowest layer (0 km) in CTRL and CS01 are almost the same (about 6.3 mm/km), but *Nw* error in CS02 at bottom layer decreases to 4.5 mm/km, with improvements of about 28% after adding ground meteorological data. At middle troposphere, for example, the 8th layer (2 km height), *Nw* error is about 3.6 mm/km in CTRL and about 3.1 mm/km in both CS01 and CS02. At upper layers, for example, the 21th layer (10 km), there are no significant differences among three cases due to small amount of water vapor, with *Nw* errors of only about 0.25 mm/km. This can be explained with the help of Figure 7 where the horizontal cross-section of

four grids at different layers are taken as examples. Compared to the middle layer (8th layer with height of 2 km) (Figure 7b), the puncture points of GNSS signals at the bottom layer for each station are more concentrated, indicating the measurements used for estimating values at the grid nodes have stronger correlations, which will result in less contributions to the estimates compared to the middle layer. Therefore, adding the GNSS observations has more significant improvements for the middle layer than the for the bottom layer. For the upper layer, due to the small amount of water vapor, the improvements are not significant although the puncture points are more discretely distributed. The introduction of ground meteorological data has strong constraints at the station location, which therefore mainly improves estimates at the bottom layers.

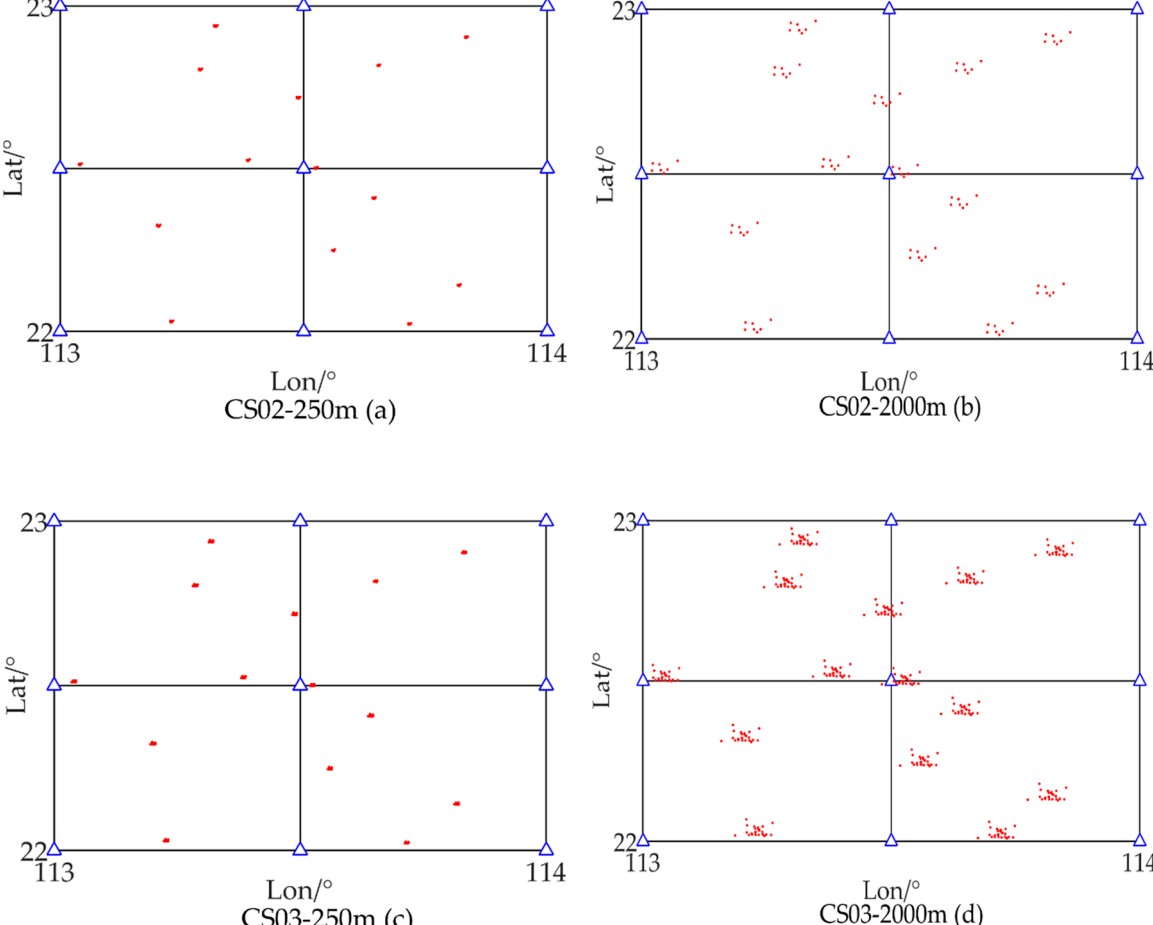

**Figure 7.** Horizontal cross-section at (**a**) 250 m layer in CS02; (**b**) 2000 m layer in CS02; (**c**) 250 m layer in CS03; (**d**) 2000 m layer in CS03. Triangles denote the to-be-estimated grid nodes. Red dots represent the puncture points of GNSS signals.

## 5.2. Influences of Multi-GNSS Observations

To test the effect of multi-GNSS observations in tropospheric tomography, two cases were carried out and compared, namely, CS02 with GPS-only SWD and CS03 with multi-GNSS SWD as described in Table 1.

Comparisons of wet refractivity error in CS02 and CS03 are presented in Figure 6 where we can find that there are only very slight improvements (about 0.7%) after the introduction of multi-GNSS observations. This may be due to the fact that the tomographic voxel is relatively flat with much wider horizontal range (0.5°) than the vertical range (12 km), resulting in negligible impacts on the percentage of voxels with signal passed (28.59% and 28.60% in CS02 and CS03, respectively). Another reason is

that the introduction of multi-GNSS has small impacts on the spatial dispersion of puncture points although the puncture points become denser as can be seen from Figure 7.

### 5.3. Influences of Ground-Based Station Distribution

To study the influence of station distribution on the tomographic results, three experiments were carried out and compared, i.e., CS04, CS05, and CS06. All the GNSS station locations in these three experiments are simulated at evenly distributed grid points with different horizontal resolution, i.e., 1°, 0.5° and 0.2° in CS04, CS05 and CS06, respectively, resulting in 45, 153, and 861 stations in the tomographic region as summarized in Table 1. The heights of the stations are interpolated from the ERA5 surface geopotential product. Comparisons of tomographic results for these three experiments are presented in Figure 8.

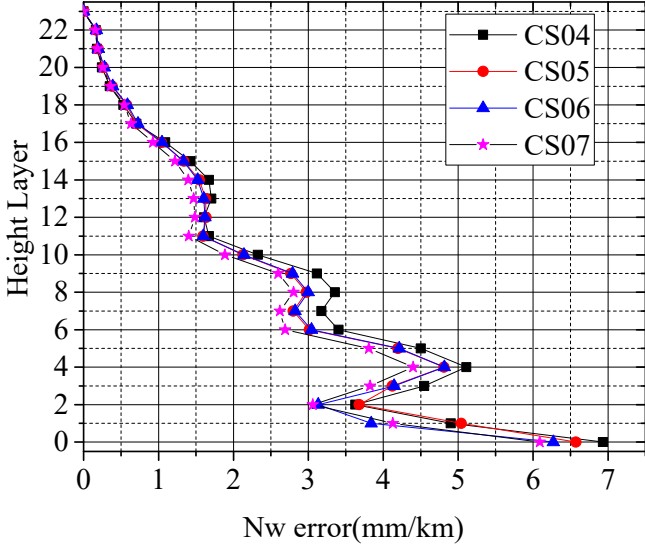

**Figure 8.** Comparisons of wet refractivity error profiles for CS04, CS05, CS06, and CS07 for a study of influences of ground-based station distribution and horizontal resolution on tomography.

Average *Nw* error is about 6.57 and 6.27 mm/km in CS05 and CS06 at the lowest layer, with improvements of about 11.2 and 15% compared to CS04, respectively. At the middle layers, for example, the 8th layer with height of 2 km, *Nw* error is almost the same in CS05 and CS06 (~3.36 mm/km), with improvement of about 11.3% compared to CS04. As can be seen in Figure 9, as the ground GNSS stations becomes denser, the puncture points of GNSS signals is generally more discretely distributed in space, which will result in better estimates of refractivity at the voxel nodes. There are almost no differences at upper layers due to the low water vapor content there. Compared to the influences of multi-GNSS observations as discussed in the previous section, we can find that increases in ground stations have larger impacts on the tomographic results than increases in GNSS observations.

### 5.4. Influences of Horizontal Resolution

Two cases (CS06 and CS07) were carried out and compared in order to study the influences of different tomographic horizontal resolution on the tomography. The horizontal resolution in CS06 and CS07 is set to be 0.5 and 0.25 degrees, resulting in 5760 and 21,528 voxels, respectively. The GNSS locations were simulated with horizontal resolution of 0.25 degrees as described in Table 1, and comparisons of wet refractivity error for CS06 and CS07 are presented in Figure 8. We can find that wet refractivity errors in CS07 are generally smaller than in CS06, especially at the middle layers (layers 4 to 10). At the lowest layer, the average *Nw* error is about 6.27 and 6.09 mm/km in CS06 and CS07, respectively, with about 3% improvement when the horizontal resolution increased from 0.5 to 0.25

degrees. At middle layers, like the 9th layer, *Nw* accuracy in CS07 is improved by about 6% compared to CS06. At heights above 18th layers, the differences are negligible because of the low humidity.

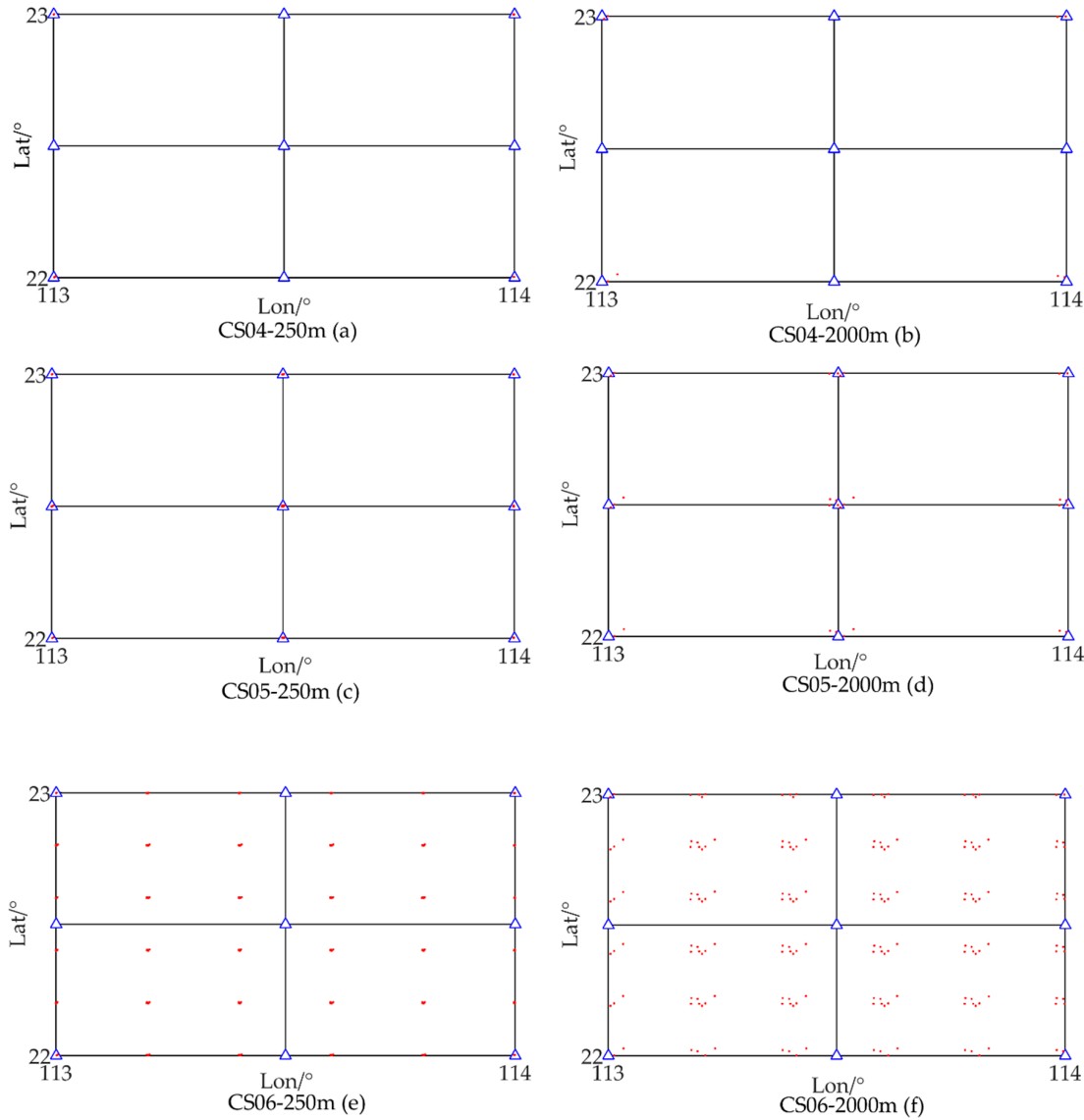

**Figure 9.** Horizontal cross-section at (**a**) 250 m layer in CS04; (**b**) 2000 m layer in CS04; (**c**) 250 m layer in CS05; (**d**) 2000 m layer in CS05; (**e**) 250 m layer in CS06; (**f**) 2000 m layer in CS06. Triangles denote the to-be-estimated grid nodes. Black dots represent the puncture points of GNSS signals.

## 6. Conclusions

Water vapor is a precursor of cloud, rain, snow, sleet, hail, and other precipitations, all of which constitute the basic phenomenon of weather as people experience every day. The water vapor distribution plays a key role in the evolution of atmospheric storm systems and vertical stability of the atmosphere. It is therefore important to get the information of atmospheric water vapor content and distribution. In recent years, GNSS including the US GPS, Russian GLONASS, EU Galileo, and Chinese Beidou has been widely employed as an operational tool for retrieving atmospheric water vapor. Compared to other water vapor measurement tools, the ground-based GNSS has the advantages of high accuracy, high temporal resolution and operations in all weather conditions. The ground-based GNSS can only provide the vertical integral of water vapor or refractivity directly. However, for some applications, the three dimensional of humidity fields are of great importance, and the tropospheric tomography technique can be used for this purpose. Near real-time tropospheric tomography can

provide useful information to weather phenomena study and navigation and positioning, but it faces challenges in rank-deficit issue and massive parameter rapid estimation which were solved by using NWP forecasting products and ART technique in this study.

The tropospheric tomographic results may be affected by many factors, such as the inclusion of ground meteorological measurements, the multi-GNSS observations, the ground-based station distribution, the tomographic resolution and so on. In order to exclude the discrepancy between the input observations (GNSS-derived slant delays) and the references (i.e., radiosondes or reanalysis products), the input observations for the tomographic system were simulated on the basis of the ERA5 reanalysis products and the reanalysis were then taken as references for the investigation of influences of different factor on tomography.

It is found that when adding GPS observation data, compared with only using GFS forecast product as background field, the tomographic results were improved by about 7%, and can be further improved by about 6% at the bottom layers when using the ground meteorological measurements.

The inclusion of multi-GNSS observations does not considerably improve the tomographic results compared with GPS-only observations. This is mainly due to the geometry of the tomographic voxel, namely, much wider in horizontal than vertical, resulting in almost unchanged ray touched voxel percentage when adding multi-GNSS observations, and the dispersion of the puncture points of the GNSS signals at different layers does not change significantly.

Finally, the effects of station density and grid partition on the results of tomography are investigated. The experiments were carried out with different stations distribution (horizontal separation of 1°, 0.5° and 0.2°) and grid horizontal resolution (0.5° and 0.25°). Results show that the $Nw$ accuracy can be further improved when the station and the grid partition become denser. However, too dense station distribution and grid partition will significantly increase the burden of tomographic solution. Therefore, in real situations, a compromise needs to be made between the accuracy and computational efficiency, especially for near real-time applications.

By excluding the discrepancy between the input observations and references, which are existed in most current studies, the investigation of influences of different factors on the tomographic results in this study can provide some indications and references for tomographic research and applications in the future.

**Author Contributions:** Writing—original draft preparation, W.L.; writing—review and editing, Y.L., W.Z.; software, J.H.; data curation, Y.Z., H.Z.

**Funding:** This work was supported by the National Key Research and Development Program of China (2016YFB0501800), the National Natural Science Foundation of China (41774036; 41804023), the China Postdoctoral Science Foundation funded project (2017M622518), and the Natural Science Foundation of Hubei Province of China (2018CFB193).

**Acknowledgments:** The authors would like to thank NOAA CISL for providing forecast products and ECMWF for providing ERA5 products.

**Conflicts of Interest:** The authors declare no conflict of interest.

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
