# Peer review of "On the Study of Influences of Different Factors on the Rapid Tropospheric Tomography"

_remotesensing, doi:10.3390/rs11131545_

Round 1
Reviewer 1 Report
A study that deserves to be continued
Author Response
Response to Reviewer 1 Comments
Point 1: A study that deserves to be continued
Response 1: We sincerely thanks for the work by Reviewer #1.

Reviewer 2 Report
Review of “On the study of influences of different factors on the rapid large-scale tropospheric tomography” by Liu et al.
I accepted the responsibility of reviewing this paper with great interest because the authors promised to provide comparative descriptions of the influences of different factors on the rapid large-scale tropospheric tomography. A better, fast and accurate GNSS water vapor tomographic method would of great contribution to the science of Geodesy, Remote Sensing and Meteorology. A well-written manuscript on this subject would be an invaluable contribution for this reason.
The authors have utilized the algebraic reconstruction technique in their study, where Slant Wet Delay (SWD) was used as an input which was derived from both GNSS processing and also from simulation (??). They have also investigated the influence of other factors such as tomographic resolution, GPS vs multi-GNSS data and so on. In each case, SWD was the input for the tomographic model.
Unfortunately, this manuscript does not provide enough detail to allow a reader to evaluate the authors' assumptions, analyses, and interpretations. In the introduction section, the authors provide a critical review previously mentioned tomographic method briefly. For example, there are some studies that concluded having improvement on the tomographic results (e.g., Dong and Jin 2018). Please compare the methods of those type of studies.
In section 2, the author should describe how SWD was derived using two methods in details. I understood that the authors have provided the details for deriving SWD from GNSS. The same level of details is needed to describe how SWD was simulated. Without this, I cannot justify the result of the simulation cases listed in Table 1. The author also mentioned that they have used PANDA, but I would recommend that they provide brief description of how GNSS data were processed. The authors should also describe how one could get STD from GNSS ZTD (shown in figure 2).
The authors might consider having a section called “Data” where they describe briefly the data that was utilized in this study, their uncertainty and so on. For example, what Global Forecasting and ERA5.
The author has to improve the writing style, such as avoiding too many complex compound sentences and also to avoid repetition. They need to be careful when describing equations. In some equation, there were some extra variables that described afterward. Such as in equation 3, there is no “mh” (hydrostatic mapping function), and also in equation 6, there is no “m” .
The author also needs to improve the structure of the paper. For example, section 3 is called “SWD Simulation” followed by section 5 “Conclusion” (There is no section 4). The reader may get the impression that this paper is all about “SWD Simulation”, which is in fact not the case.
I have provided comment up to line 235. Unfortunately, there is not much I could contribute as a reviewer by reading further because of the inadequacy of information provided up to that point.
I strongly encourage all authors of this manuscript to read it over carefully, to check it for accuracy and to rewrite the whole manuscript prior to resubmission. Where additional details are requested, they can be provided either in the main text or in a supplement -- but this manuscript simply won't be publishable without substantial clarification and additional information.
Specific Comments:
Abstract:
Line 13 - 16: Rewrite the first sentence of the abstract. Emphasize the “influences of the different factors” rather than “the rapid large-scale tropospheric tomography system”, since the whole paper discussed the effect of different factors. Reduce the use of complex compound sentence.
Line 16 – 19: Please re-write this sentence for clarity. One would get the impression that throughout the study ONLY the simulated observations was used as input, although later in the manuscript it was described that Slant Wet Delay (SWD) from GNSS processing was used. I would recommend describing method here explicitly with one or two sentences, since the following line started to describe results.
Line 19 – 21: Readers may find this sentence very confusing. SWD is the input for this tomography experiment. One might get the impression that adding SWD is causing the improvement. Or is it saying that adding meteorological data causes improvement?
Authors claimed to have an improvement in the humidity field. Readers may find it confusing, since it may not have been in the manuscript. Also, what is the range of the middle and bottom layer?
Line 25 – 26: Is it the resolution of the tomography field?
Line 27 – 32: This section seems to be a reiteration of the result section above. Such as, line 29 – “Adding ground meteorological … ” is a repetition of line 20. Line 29 – “But multi GNSS observation … ” is a repetition of line 21 – 22.
Section 1: Introduction:
Line 63 – 65: Author should expand on this if any study is done before on large scale rapid tomography. If yes, discuss some those briefly and cite.
Line 66 – 67: Authors should expand on this. Have the influences of different factors on the tomographic result already been studied as author cited some previous studies here? If it is, please describe what is different for this study?
Line 72 – 73: The reader may find it confusing. Isn’t the slant wet delay ALSO obtained from GNSS derived zenith wet delay as described in section 2 and Table 1? What does “simulated” refer to? Is it calculated using a method that does not require “GNSS derived ZWD”? If that is the case, then explain that method somewhere in section 2?
Line 73 – 75: What is the “reference” here?
Section 2: GNSS Tropospheric Tomography Methodology
This section needs a subsection on how the SWD was simulated. Since this paper is using SWD from two sources such as from GNSS ZWD and from ray tracing using ERA5 weather model. I would recommend that the authors discuss the second method hereafter section 2.1. The authors can move section 3.1 and 3.1 here and provide more details, instead of just saying that SWD was simulated.
Line 89: Add “(ZTD)” after Zenith Tropospheric Delay since this is the first instance.
Line 100: “mh” is not in equation 3.
Line 101 – 102: Epsilon should be elevation angle and phi should be azimuth.
Section 2.2: Tomography Parameter Estimation Method
In this section, the author should summarize other techniques used in tropospheric tomography and then justify why this method is better.
Line 132: Where is “m” in equation 6?
3. SWD Simulations
Please see my comments from section 2.
Line 186: Replace “slat” with “slant”.
Line 195 - 197: The section heading says that SWD was simulated. Why was this line mentioned “SWD or STD”? Were they both estimated using ray tracing? If that is the case use “and” instead of “or”.
Line 197 - 199: Authors used GNSS-derived Slant Total Delay (STD) for evaluation. They should describe in section 2 how one would estimate STD from GNSS analysis.
Line 199 – 200: What does it mean by “meteorological data were simulated” from ERA5? Were the pressure, temperature and relative humidity extracted for the location of GNSS or along the ray path?
Line 233 – 235: Please re-write this sentence for clarity. Please clarify what it means by “meteorological observations ….. were simulated …. ”. Also, there is no SWD in figure 3.
Reference:
Dong Z, Jin S (2018) 3-D Water Vapor Tomography in Wuhan from GPS, BDS and GLONASS Observations. Remote Sensing 10:62. doi: 10.3390/rs10010062
Author Response
Response to Reviewer 1 Comments
General Comments
I accepted the responsibility of reviewing this paper with great interest because the authors promised to provide comparative descriptions of the influences of different factors on the rapid large-scale tropospheric tomography. A better, fast and accurate GNSS water vapor tomographic method would of great contribution to the science of Geodesy, Remote Sensing and Meteorology. A well-written manuscript on this subject would be an invaluable contribution for this reason.
The authors have utilized the algebraic reconstruction technique in their study, where Slant Wet Delay (SWD) was used as an input which was derived from both GNSS processing and also from simulation (??). They have also investigated the influence of other factors such as tomographic resolution, GPS vs multi-GNSS data and so on. In each case, SWD was the input for the tomographic model.
Point 1: Unfortunately, this manuscript does not provide enough detail to allow a reader to evaluate the authors' assumptions, analyses, and interpretations. In the introduction section, the authors provide a critical review previously mentioned tomographic method briefly. For example, there are some studies that concluded having improvement on the tomographic results (e.g., Dong and Jin 2018). Please compare the methods of those type of studies.
Response 1: We have added the details of the relevant references and the article (Dong and Jin, 2018) recommended by the reviewer have also been referred.
Point 2: In section 2, the author should describe how SWD was derived using two methods in details. I understood that the authors have provided the details for deriving SWD from GNSS. The same level of details is needed to describe how SWD was simulated. Without this, I cannot justify the result of the simulation cases listed in Table 1.
Response 2: We added section 4.1 “Simulation Procedure” to describe how SWD was simulated.
Point 3: The author also mentioned that they have used PANDA, but I would recommend that they provide brief description of how GNSS data were processed.
Response 3: The process of GNSS data have been briefly description in section 4.2:
“The ERPs (Earth Rotation Parameters) and the DCBs (Differential Code Biases) provided by CODE were employed. The absolute antenna phase center, phase windup corrections and station displacement corrections suggested in the IERS 2003 conventions were applied. The elevation cut-off angle was 7 and an elevation-dependent weighting strategy was applied to measurements at low elevations. The Global mapping function was used to relate zenith tropospheric delays (ZTDs) to the measurements. From a selected CORS network, the satellite clocks are estimated firstly as white noise with the satellite orbits and the station coordinates fixed and then the satellite orbit and the estimated clocks are applied to user station for kinematic PPP in post-mission mode. ZTDs are estimated as piece-wise constants with a constraint of 4 cm2 per hour and ambiguities are estimated as float solutions.”
Point 4: The authors should also describe how one could get STD from GNSS ZTD (shown in figure 2).
Response 4: In the section 3.1 of the new manuscript, we have described how could STD be obtained. Firstly, ZHD can be estimated by Hopfield or Saastamoniene model. Then, according to equation 3, ZWD could be obtained by ZTD minus ZHD. Finally, STD could be got by equation 4.
Point 5: The authors might consider having a section called “Data” where they describe briefly the data that was utilized in this study, their uncertainty and so on. For example, what Global Forecasting and ERA5.
Response 5: We have added section 2 to describe the various data used, including: GNSS data, Reanalysis product, NWP forecast product.
Point 6: The author has to improve the writing style, such as avoiding too many complex compound sentences and also to avoid repetition. They need to be careful when describing equations. In some equation, there were some extra variables that described afterward. Such as in equation 3, there is no “mh” (hydrostatic mapping function), and also in equation 6, there is no “m” .
Response 6: It has been removed.
Point 7: The author also needs to improve the structure of the paper. For example, section 3 is called “SWD Simulation” followed by section 5 “Conclusion” (There is no section 4). The reader may get the impression that this paper is all about “SWD Simulation”, which is in fact not the case.
Response 7: The structure of the paper has been adjusted as you said.
Point 8: I have provided comment up to line 235. Unfortunately, there is not much I could contribute as a reviewer by reading further because of the inadequacy of information provided up to that point.
Response 8: According to your comments, we added a section 2 called “Data”, related equations, more references and so on to help reader to understand this manuscript.
Point 9: I strongly encourage all authors of this manuscript to read it over carefully, to check it for accuracy and to rewrite the whole manuscript prior to resubmission. Where additional details are requested, they can be provided either in the main text or in a supplement -- but this manuscript simply won't be publishable without substantial clarification and additional information.
Response 9: Thanks for your comment, according to your comments, the entire manuscript has undergone major adjustments from structure to content.
Specific Comments:
Abstract:
Point 10: Line 13 - 16: Rewrite the first sentence of the abstract. Emphasize the “influences of the different factors” rather than “the rapid large-scale tropospheric tomography system”, since the whole paper discussed the effect of different factors. Reduce the use of complex compound sentence.
Response 10: We have deleted the “large-scale” from the first sentence and rewrite the sentence. And the title of the manuscript has also been changed.
Point 11: Line 16 – 19: Please re-write this sentence for clarity. One would get the impression that throughout the study ONLY the simulated observations was used as input, although later in the manuscript it was described that Slant Wet Delay (SWD) from GNSS processing was used. I would recommend describing method here explicitly with one or two sentences, since the following line started to describe results.
Response 11: It has been changed:
“In order to exclude the impacts from discrepancies of water vapor information between input observations and references on the tomographic results, the latest reanalysis products, ERA5, which were taken as references for result evaluations, were used to simulate slant wet delay (SWD) observations at GNSS stations. Besides, the slant delays derived from GNSS processing were also used to evaluate the reliability of simulated observations.”
Point 12: Line 19 – 21: Readers may find this sentence very confusing. SWD is the input for this tomography experiment. One might get the impression that adding SWD is causing the improvement. Or is it saying that adding meteorological data causes improvement?
Authors claimed to have an improvement in the humidity field. Readers may find it confusing, since it may not have been in the manuscript. Also, what is the range of the middle and bottom layer?
Response 12: Actually, tomography results show that the input both SWD and ground meteorological data could improve the tomographic results where SWD mainly improve the results at middle layers (500 to 5000m, namely 2 to 16 layer) and ground meteorological data could improve the humidity fields at bottom layers further (0 to 500m, namely 0 to 2 layer).
Point 13: Line 25 – 26: Is it the resolution of the tomography field?
Response 13: Yes, 0.5 and 0.25 are the horizontal resolution of tomography field.
Point 14: Line 27 – 32: This section seems to be a reiteration of the result section above. Such as, line 29 – “Adding ground meteorological … ” is a repetition of line 20. Line 29 – “But multi GNSS observation … ” is a repetition of line 21 – 22.
Response 14: We have deleted this section as you said.
Section 1: Introduction:
Point 15: Line 63 – 65: Author should expand on this if any study is done before on large scale rapid tomography. If yes, discuss some those briefly and cite.
Response 15: We have added the related references:
“Most of previous troposphere tomography studies aimed at small regions generally in the scale of several tens of kilometers with relatively dense GNSS networks [2, 8]. There have been few attentions paid to the rapid tomography over wide area. However, rapid tropospheric tomography over wide area can be very useful in some applications, such as providing timely and complete pictures for some extreme weather phenomena, supporting the development of high-resolution mapping functions, and providing consistent tropospheric delay corrections for (near) real-time precise navigation or positioning users over wide area.”
Point 16: Line 66 – 67: Authors should expand on this. Have the influences of different factors on the tomographic result already been studied as author cited some previous studies here? If it is, please describe what is different for this study?
Response 16: We have expanded it as you comment: (line 72 - 82)
“The tomographic results can be affected by many factors. For example, Yu et al. [9] compared various constraint conditions. As the results shown, …”
Point 17: Line 72 – 73: The reader may find it confusing. Isn’t the slant wet delay ALSO obtained from GNSS derived zenith wet delay as described in section 2 and Table 1? What does “simulated” refer to? Is it calculated using a method that does not require “GNSS derived ZWD”? If that is the case, then explain that method somewhere in section 2?
Response 17: In this manuscript, we used two kinds of ZWD, which are measured ZWD calculated by PANDA software (high precision GNSS data processing software developed by Wuhan University) and simulated ZWD obtained from the method described in Section 4.1. It should be pointed out that the measured is mainly used to evaluate the accuracy of the simulated. But when verifying the accuracy of simulation ZWD, ZWD is not used directly for evaluation, but STD is used for indirect evaluation.
Point 18: Line 73 – 75: What is the “reference” here?
Response 18: The wet refractivity filed derived from the tomography will be evaluated by comparing with reanalysis products, ERA5 (as a true value), to exclude the influence of differences between inputs and references.
Section 2: GNSS Tropospheric Tomography Methodology
Point 19: This section needs a subsection on how the SWD was simulated. Since this paper is using SWD from two sources such as from GNSS ZWD and from ray tracing using ERA5 weather model. I would recommend that the authors discuss the second method hereafter section 2.1. The authors can move section 3.1 and 3.1 here and provide more details, instead of just saying that SWD was simulated.
Response 19: We have added section 4.1 named “Simulation Procedure” to describe how the SWD was simulated.
Point 20: Line 89: Add “(ZTD)” after Zenith Tropospheric Delay since this is the first instance.
Response 20: It has been changed as you said.
Point 21: Line 100: “mh” is not in equation 3.
Response 21: It has been removed.
Point 22: Line 101 – 102: Epsilon should be elevation angle and phi should be azimuth.
Response 22: It has been changed as you said.
Section 2.2: Tomography Parameter Estimation Method
Point 23: In this section, the author should summarize other techniques used in tropospheric tomography and then justify why this method is better.
Response 23: The commonly used tomography algorithms can be divided into two categories: iterative reconstruction algorithm and non-iterative reconstruction algorithm. Non-iterative reconstruction algorithm includes Kalman filter and so on, but considering the running ability of the computer, it is not suitable for large matrix operation, especially for inverse operation. In this study, the algebraic reconstruction techniques (ART) in iterative reconstruction algorithm will be utilized…
Point 24: Line 132: Where is “m” in equation 6?
Response 24: It has been removed.
3. SWD Simulations
Please see my comments from section 2.
Point 25: Line 186: Replace “slat” with “slant”.
Response 25: It has been changed as you said.
Point 26: Line 195 - 197: The section heading says that SWD was simulated. Why was this line mentioned “SWD or STD”? Were they both estimated using ray tracing? If that is the case use “and” instead of “or”.
Response 26: It has been changed to “and” as you said.
Point 27: Line 197 - 199: Authors used GNSS-derived Slant Total Delay (STD) for evaluation. They should describe in section 2 how one would estimate STD from GNSS analysis.
Response 27: We have added equation 4 to describe how to obtain STD.
Point 28: Line 199 – 200: What does it mean by “meteorological data were simulated” from ERA5? Were the pressure, temperature and relative humidity extracted for the location of GNSS or along the ray path?
Response 28: Actually, the meteorological data refer to the ground meteorological observation data, namely, the pressure, temperature and relative humidity at ground GNSS stations. Because we cannot get the meteorological observation data at the actual station, the meteorological observation data are simulated on the basis of ERA5 reanalysis data.
Point 29: Line 233 – 235: Please re-write this sentence for clarity. Please clarify what it means by “meteorological observations ….. were simulated …. ”. Also, there is no SWD in figure 3.
Response 29: It has been changed:
“SWD at 93 GNSS stations in Guangdong Province on the day 119 in 2017 and the meteorological observations (air temperature, pressure and relative humidity) at GNSS station were simulated on the basis of ERA5.”
Reference:
Dong Z, Jin S (2018) 3-D Water Vapor Tomography in Wuhan from GPS, BDS and GLONASS Observations. Remote Sensing 10:62. doi: 10.3390/rs10010062

Reviewer 3 Report
Journal: Remote Sensing (ISSN 2072-4292)
Manuscript ID: remotesensing-500988
Title: "On the study of influences of different factors on the rapid tropospheric tomography"
Authors: Wenxuan Liu, Yidong Lou, Weixing Zhang, Jinfang Huang, Yaozong Zhou, Haoshan Zhang
General Comments to the authors:
In this study, the authors propose to evaluate the influence of different factors on the rapid tropospheric tomography, such as the inclusion of ground meteorological measurements, the multi-GNSS observations, the ground-based station distribution, tomographic resolution. The input observations for the tomographic system were simulated on the basis of the ERA5 reanalysis products and the reanalysis was then taken as references for the investigation of influences of the different factor on tomography. In my opinion, this study is very interesting for readers of the Remote Sensing Journal. However, the current version of the manuscript needs to be improved before to be published, because the presented results are obvious (not present additional arguments that it is known) and the rapid tomography results in real application are not presented, in which (in my opinion) the influence of the different factors should be discussed. This study appears to be fragmented, and an important part was not presented in the current version of the paper. The analysis of the result is poor, which should be better organized. There are several scientific points to improve the manuscript, which deserve the attention of the authors (for more details see specific comments):
- Presentation: The bibliographic revision need to be improved and many important previous works should be included. There are some very short sections, which can be avoided. The English language and style are good, but some few mistakes should be corrected. Some equation and figure need to be improved. There are some repeated ideas in the manuscript, which should be rewritten;
- Data used: I don’t know if only one day is enough to evaluate the influence of different factors on the rapid tropospheric tomography. I think that it will be interesting to see the influence of these factors in the tropospheric tomography during extreme events
- Results: The results presented in section 4.2 about the discrepancy between the simulated STD and GNSS-derived STD should be removed from this manuscript because this discrepancy has no impacts on the tomographic results. It is interesting to access the quality of simulated SMD and STD based on ERA5, but it is another study.
- Analysis of results: The influence of the different factors on the rapid tropospheric tomography is done exploring only the vertical profile and horizontal cross-section. This study is poor, which should be better presented. Some results about the influence of these factors during extreme events could help in this topic. Besides, other aspects should be taken into consideration in this analysis. For example, the quality of the humidity tomography, the horizontal distribution of the error, the spatial detailing of the humidity, and computational burden and time spend. The results presented can be organized and only two figures, which is not enough to compose the paper. In my opinion, this study needs to be reorganized and evaluation of these factors should be done during weather extreme events.
Specific comments to the authors:
Introduction
Line 43 and many others – The bibliographic revision presented in the introduction is very poor and many important previous works are not mentioned and there are many statements that should be put the reference. I suggest reading the introduction of this paper available at https://doi.org/10.1002/met.1735 and put the reference in the lines: 43, 45, 51, 84 and other section of the manuscript, for example in line 139, and others.
Lines 87-95: The objective of the paper is not suitably elaborated and present the research fragmented. Please see my comment in the next sections.
Lines 89 and many others: The English language and style are good, and I could not find orthographic mistakes, besides I am not native English and I am not able to suitable evaluate this topic. However, I identify an equivocal usage of the verb “to be” in the future tense “will be” to describe what the reader finds in the subsequent section in the manuscript. In my opinion, this is a style questionable to be used in a scientific document, because it is not future in the time, but in the sequence used to present the information in the manuscript. I suggest changing “will be” to “is” in lines 89, 92. 93, 94, 95 96, 97, 98, 100, 103, 106, 170, 230, 231, 241, 253, 307 and many other lines. Please, consult a native speaker reviser about this.
Section 1: Data
Line 106 – Why was this day (DoY 119) chosen? Does this day characterize a typical atmospheric behavior? Is only one day enough to evaluate the influence of different factors on the rapid tropospheric tomography? I think that it will be interesting to see the influence of these factors in the tropospheric tomography during extreme events, as is mentioned by authors in the introduction.
Section 3: GNSS Tomography Methodology
Line 135 – Change the term “eliminated” by “minimized”.
Line 141 and 145– The equation (2) and (3) are the same. Please avoid the prolix text.
Line 148 – The term mw is erroneously repeated in equation (4). See my suggestion for the new equation 4 in the next item.
Line 148 and 154– The second term of the equation (4) and (5) are the same. Please avoid the prolix text, again. Please, change the equation (4) to:
STD= SHD+ SWD= mh ZHD+SWD (4)
and maintain the equation (5), as was presented.
Lines 162 -165 – This paragraph expresses the same idea using the same work to it was presented in the lines from 66 to 70. Please, rewrite this.
Line 142 and others – please, remove the space at the beginning of the line after the equation (2), because is the same phase. The same is observed in lines 149, 159, 187, 193, 213 and 233.
Line 199 – Avoid the very short section, which has only one paragraph. The same is observed in subsection 3.3 (line 215).
Section 4
Line 258 - 262: The authors state that the discrepancy between the simulated STD and GNSS-derived STD has no impacts on the tomographic results. I agree with this, but I do not understand the reason for this evaluation is presented in this study. It is interesting to access the quality of simulated SMD and STD based on ERA5, but it is another study and should be removed from the manuscript.
Line 264 – Figure 3 should be presented after this paragraph.
Line 281 – What does “Satellite altitude angle” mean? I think that the author would like to say “Satellite elevation angle”. Please, correct this. The same in figure 5.
Line 283 – Why were G01 and G010 chosen to be evaluated in figure 4? In figure 5, it is possible to see that these stations present the lower RMS, and consequently, they do not represent the general behavior. The authors need to pay more attention because these results sub estimate the real values. I thank that a better evaluation should be presented here, using many other stations and other statistics, for example in percentage term, etc.
Line 285-290 – The analysis of the results are superficial and not explore the information provide for figures and not explains the results presented for them. What the reason for different stations present different RMS? The figure about RMS in the function of Satellite elevation angle, the more interesting is the RMS for elevation angle below 10°, which are not discussed by authors.
Section 5
Line 312 - I suggest present the results about the tropospheric tomography over the GNSS network present in figure 1, particularly during the weather extreme events, which the influence of the different factors could be evaluated. The tomography results generated using the methodology reported here in real time is not explored in this paper.
Line 312 – The number of experiments is larger than necessary. I do not find a reason to present the case 05
Lines 313, 349 363, 388- the analyses of the result are very simplified. Only the vertical profile and horizontal cross-section are explored. This study is poor, which should be better presented. Some results about the influence of these factors during extreme events could help in this topic. Besides, other aspects should be taken into consideration in this analysis. For example, the quality of the humidity tomography, the horizontal distribution of the error, the spatial detailing of the humidity (particularly in the subsection 5.4), and computational burden and time spend, which is very important to real-time application in a real situation of extreme events.
Lines 327, 354, 372 and 401– Figure 6 and 8 should compose the one unique plot and figure 9 and 11 should another plot of the same figure. The analysis of results could be separated in sub-sections. As these plots are vertical profile, I suggest change the axis x for y, consequently the “Height layers” would be presented in the vertical direction in these figures, make easier the interpretation of results. Please put the unity [km] in the “Height Layer” label.
Lines 346 and 385 – I strongly suggest improve figure 7 and 10. I think that is possible to integrate this information in a new more efficient and intelligent figure. It is impossible to see triangles and dots in figure 7 and in figure 10 there is empty space.
Lines 327, 354, 372 and 401: In this figure, it is observed an abnormality in the behavior of error in the function of the height in vertical profile, which a pick is observed in 8 km in the figures 6 and 8 and 4 km in the figures 9 e 11. The authors did not discuss these results and present the reason for this behavior. Please correct this topic.
Section 5: Conclusions
Lines 404 -421: In this part of the manuscript there are many repeated ideas explored in another part of the text. This part needs to be enriched with other additional arguments to discuss the importance of this study.
Lines 422 and after: The conclusion reported state that meteorological stations are important, in special lower layers, a denser network of the receiver has more influence in the troposphere tomography than a denser network of the satellite, and the better horizontal resolution improves the quality of tomography products. These conclusions are obvious and the results presented are not necessary to prove them. In my opinion, this study needs to be reorganized and evaluation of these factors should be done during weather extreme events.
Author Response
Response to Reviewer 3 Comments
Point 1: Presentation: The bibliographic revision need to be improved and many important previous works should be included. There are some very short sections, which can be avoided. The English language and style are good, but some few mistakes should be corrected. Some equation and figure need to be improved. There are some repeated ideas in the manuscript, which should be rewritten.
Response 1: Thanks for your comments, we have changed the manuscript as you said. Please refer to the following for details.
Point 2: Data used: I don’t know if only one day is enough to evaluate the influence of different factors on the rapid tropospheric tomography. I think that it will be interesting to see the influence of these factors in the tropospheric tomography during extreme events.
Response 2: Although we only calculated the data for one day, we solved it every half hour, that is to say, we calculated 48 times and finally took the average. So, the result is still statistically significant to a certain extent. As for the extreme events, the selected day is before the heavy rain, so the water vapor also has an obvious change.
Point 3: Results: The results presented in section 4.2 about the discrepancy between the simulated STD and GNSS-derived STD should be removed from this manuscript because this discrepancy has no impacts on the tomographic results. It is interesting to access the quality of simulated SMD and STD based on ERA5, but it is another study.
Response 3: What we're evaluating here is the method of simulation, not the ERA5 data. So, we think it's necessary to add this section.
Point 4: Analysis of results: The influence of the different factors on the rapid tropospheric tomography is done exploring only the vertical profile and horizontal cross-section. This study is poor, which should be better presented. Some results about the influence of these factors during extreme events could help in this topic. Besides, other aspects should be taken into consideration in this analysis. For example, the quality of the humidity tomography, the horizontal distribution of the error, the spatial detailing of the humidity, and computational burden and time spend. The results presented can be organized and only two figures, which is not enough to compose the paper. In my opinion, this study needs to be reorganized and evaluation of these factors should be done during weather extreme events.
Response 4: We have added the computational burden and computational efficiency in table 1.
Point 5: Line 43 and many others – The bibliographic revision presented in the introduction is very poor and many important previous works are not mentioned and there are many statements that should be put the reference. I suggest reading the introduction of this paper available at https://doi.org/10.1002/met.1735 and put the reference in the lines: 43, 45, 51, 84 and other section of the manuscript, for example in line 139, and others.
Response 5: We have added the reference as you said.
Point 6: Lines 87-95: The objective of the paper is not suitably elaborated and present the research fragmented. Please see my comment in the next sections.
See our responses below in the next sections.
Response 6: Thanks for your comments.
Point 7: Lines 89 and many others: The English language and style are good, and I could not find orthographic mistakes, besides I am not native English and I am not able to suitable evaluate this topic. However, I identify an equivocal usage of the verb “to be” in the future tense “will be” to describe what the reader finds in the subsequent section in the manuscript. In my opinion, this is a style questionable to be used in a scientific document, because it is not future in the time, but in the sequence used to present the information in the manuscript. I suggest changing “will be” to “is” in lines 89, 92. 93, 94, 95 96, 97, 98, 100, 103, 106, 170, 230, 231, 241, 253, 307 and many other lines. Please, consult a native speaker reviser about this.
Response 7: They have been changed as you said.
Point 8: Line 106 – Why was this day (DoY 119) chosen? Does this day characterize a typical atmospheric behavior? Is only one day enough to evaluate the influence of different factors on the rapid tropospheric tomography? I think that it will be interesting to see the influence of these factors in the tropospheric tomography during extreme events, as is mentioned by authors in the introduction.
Response 8: Although we only calculated the data for one day, we solved it every half hour, that is to say, we calculated 48 times and finally took the average. So, the result is still statistically significant to a certain extent. As for the extreme events, the selected day is before the heavy rain, so the water vapor also has an obvious change.
Point 9: Line 135 – Change the term “eliminated” by “minimized”.
Response 9: It has been changed as you said.
Point 10: Line 141 and 145– The equation (2) and (3) are the same. Please avoid the prolix text.
Response 10: Yes, equation (2) and (3) are a different expression of the same equation. But for the convenience of understanding, we think it is better to keep them.
Point 11: Line 148 – The term mw is erroneously repeated in equation (4). See my suggestion for the new equation 4 in the next item.
Response 11: It has been changed as you said.
Point 12: Line 148 and 154– The second term of the equation (4) and (5) are the same. Please avoid the prolix text, again. Please, change the equation (4) to:
STD= SHD+ SWD= mh ZHD+SWD (4)
and maintain the equation (5), as was presented.
Response 12: It has been changed as you said.
Point 13: Lines 162 -165 – This paragraph expresses the same idea using the same work to it was presented in the lines from 66 to 70. Please, rewrite this.
Response 13: It have been changed. “Rapid reconstructions of water vapor fields over wide area are meaning for providing high temporal and spatial humidity fields, improving the accuracy of numerical weather forecast, et al.”
Point 14: Line 142 and others – please, remove the space at the beginning of the line after the equation (2), because is the same phase. The same is observed in lines 149, 159, 187, 193, 213 and 233.
Response 14: They have been removed.
Point 15: Line 199 – Avoid the very short section, which has only one paragraph. The same is observed in subsection 3.3 (line 215).
Response 15: We merged and modified the shorter chapters.
Point 16: Line 258 - 262: The authors state that the discrepancy between the simulated STD and GNSS-derived STD has no impacts on the tomographic results. I agree with this, but I do not understand the reason for this evaluation is presented in this study. It is interesting to access the quality of simulated SMD and STD based on ERA5, but it is another study and should be removed from the manuscript.
Response 16: What we're evaluating here is the method of simulation, not the ERA5 data. So, we think it's necessary to add this section.
Point 17: Line 264 – Figure 3 should be presented after this paragraph.
Response 17: It has been changed as you said.
Point 18: Line 281 – What does “Satellite altitude angle” mean? I think that the author would like to say “Satellite elevation angle”. Please, correct this. The same in figure 5.
Response 18: It has been changed as you said.
Point 19: Line 283 – Why were G01 and G010 chosen to be evaluated in figure 4? In figure 5, it is possible to see that these stations present the lower RMS, and consequently, they do not represent the general behavior. The authors need to pay more attention because these results sub estimate the real values. I thank that a better evaluation should be presented here, using many other stations and other statistics, for example in percentage term, etc.
Response 19: G01 and G10 are selected as examples, and the assessment of all satellites are shown below in figure 5.
Point 20: Line 285-290 – The analysis of the results are superficial and not explore the information provide for figures and not explains the results presented for them. What the reason for different stations present different RMS? The figure about RMS in the function of Satellite elevation angle, the more interesting is the RMS for elevation angle below 10°, which are not discussed by authors.
Response 20: Figure 5 is the RMS of the STD differences between simulated and GNSS STD with GPS satellite and satellite elevation angle. There is no reference to stations. Besides, The transverse coordinates of the right image of figure 5 are the range of satellite elevation angles, which have low elevation angles ranging from 0° to 10°.
Point 21: Line 312 - I suggest present the results about the tropospheric tomography over the GNSS network present in figure 1, particularly during the weather extreme events, which the influence of the different factors could be evaluated. The tomography results generated using the methodology reported here in real time is not explored in this paper.
Response 21: The selected day is before the heavy rain, so the water vapor also has an obvious change. And we added “Computational Burden and Cost Time” in table 1, which reflects the needs of real time to some extent.
Point 22: Line 312 – The number of experiments is larger than necessary. I do not find a reason to present the case 05
Response 22: With one more group of experiment, the results are more statistical. As for section 5.4, influence of horizontal resolution, only two groups of experiments were taken that is because the horizontal resolution of 0.25° is enough for tomographic experiment.
Point 23: Lines 313, 349 363, 388- the analyses of the result are very simplified. Only the vertical profile and horizontal cross-section are explored. This study is poor, which should be better presented. Some results about the influence of these factors during extreme events could help in this topic. Besides, other aspects should be taken into consideration in this analysis. For example, the quality of the humidity tomography, the horizontal distribution of the error, the spatial detailing of the humidity (particularly in the subsection 5.4), and computational burden and time spend, which is very important to real-time application in a real situation of extreme events.
Response 23: We have added the computational burden and cost time in table 1. “Computational Burden and Cost Time” means the time required to calculate half an hour data (The tomographic results are output every half an hour) (PC configuration: CPU, Intel Core i7 – 7700 @ 3.60 GHz, 3.60 GHz; RAM, 8G).
Point 24: Lines 327, 354, 372 and 401– Figure 6 and 8 should compose the one unique plot and figure 9 and 11 should another plot of the same figure. The analysis of results could be separated in sub-sections. As these plots are vertical profile, I suggest change the axis x for y, consequently the “Height layers” would be presented in the vertical direction in these figures, make easier the interpretation of results. Please put the unity [km] in the “Height Layer” label.
Response 24: The figures have been changed as you said.
Point 25: Lines 346 and 385 – I strongly suggest improve figure 7 and 10. I think that is possible to integrate this information in a new more efficient and intelligent figure. It is impossible to see triangles and dots in figure 7 and in figure 10 there is empty space.
Response 25: First of all, thank you for your comments, but we do not quite understand the meaning of your comment, perhaps you can further express your views.
Point 26: Lines 327, 354, 372 and 401: In this figure, it is observed an abnormality in the behavior of error in the function of the height in vertical profile, which a pick is observed in 8 km in the figures 6 and 8 and 4 km in the figures 9 e 11. The authors did not discuss these results and present the reason for this behavior. Please correct this topic.
Response 26: Because GFS forecast product is taken as initial values in tomography and the latest reanalysis products, ERA5, which were taken as references for result evaluations, the abnormality may be caused by the systematic difference between them.
Point 27: Lines 404 -421: In this part of the manuscript there are many repeated ideas explored in another part of the text. This part needs to be enriched with other additional arguments to discuss the importance of this study.
Response 27: It have been changed as you said.
Point 28: ·Lines 422 and after: The conclusion reported state that meteorological stations are important, in special lower layers, a denser network of the receiver has more influence in the troposphere tomography than a denser network of the satellite, and the better horizontal resolution improves the quality of tomography products. These conclusions are obvious and the results presented are not necessary to prove them. In my opinion, this study needs to be reorganized and evaluation of these factors should be done during weather extreme events.
Response 28: Thanks for your comments, for the situation of extreme weather, it is the next research goal of this paper.

Round 2
Reviewer 2 Report
The manuscript has been improved substantially from the original submission. However, some minor issues remain.
Line 22: use "improves" or "could improve.
Equation 4: Mapping function for ZHD would be "mh".
Author Response
Point 1: Line 22: use "improves" or "could improve.
Response 1: It has been changed as you said.
Point 2: Equation 4: Mapping function for ZHD would be "mh".
Response 2: It has been changed as you said.
